# NMDA and AMPA Receptor Autoantibodies in Brain Disorders: From Molecular Mechanisms to Clinical Features

**DOI:** 10.3390/cells10010077

**Published:** 2021-01-05

**Authors:** Fabrizio Gardoni, Jennifer Stanic, Diego Scheggia, Alberto Benussi, Barbara Borroni, Monica Di Luca

**Affiliations:** 1Department of Pharmacological and Biomolecular Sciences, University of Milan, 20133 Milan, Italy; jennifer.stanic@unimi.it (J.S.); diego.scheggia@unimi.it (D.S.); monica.diluca@unimi.it (M.D.L.); 2Neurology Unit, Centre for Neurodegenerative Disorders, Department of Clinical and Experimental Sciences, University of Brescia, 25123 Brescia, Italy; alberto.benussi@unibs.it (A.B.); barbara.borroni@unibs.it (B.B.)

**Keywords:** autoimmunity, glutamate, brain disorders, excitatory synapse

## Abstract

The role of autoimmunity in central nervous system (CNS) disorders is rapidly expanding. In the last twenty years, different types of autoantibodies targeting subunits of ionotropic glutamate receptors have been found in a variety of patients affected by brain disorders. Several of these antibodies are directed against NMDA receptors (NMDAR), mostly in autoimmune encephalitis, whereas a growing field of research has identified antibodies against AMPA receptor (AMPAR) subunits in patients with different types of epilepsy or frontotemporal dementia. Several in vitro and in vivo studies performed in the last decade have dramatically improved our understanding of the molecular and functional effects induced by both NMDAR and AMPAR autoantibodies at the excitatory glutamatergic synapse and, consequently, their possible role in the onset of clinical symptoms. In particular, the method by which autoantibodies can modulate the localization at synapses of specific target subunits leading to functional impairments and behavioral alterations has been well addressed in animal studies. Overall, these preclinical studies have opened new avenues for the development of novel pharmacological treatments specifically targeting the synaptic activation of ionotropic glutamate receptors.

## 1. Introduction

Autoimmunity is an emerging field of research that can potentially play a key role in a better understanding of different types of Central Nervous System (CNS) disorders [1,2,3,4,5]. Autoantibodies that target neuronal surface proteins, including neurotransmitter receptors, have been described mainly in association with autoimmune encephalitis, which prominently features psychiatric symptoms, cognitive impairment, and seizures [6]. However, the potential role of these autoantibodies in different forms of chronic disorders, such as epilepsy, schizophrenia, and dementia, is of increasing interest [1,3]. Autoantibodies directed against subunits of both N-methyl-D-aspartate receptor (NMDAR) and α-amino-3-hydroxy-5-methyl-4-isoxazolepropionic acid receptor (AMPAR) seem to be among the most diffuse, abundant, and clinically relevant autoimmune antibodies identified in the last decades [4]. We will review clinical and preclinical studies that have hallowed the characterization of most of their pathological roles in the brain, and the identification of different molecular mechanisms by which they can affect the synaptic localization and function of the target receptor and, consequently, impair excitatory signaling in affected brain circuits (Table 1).

## 2. NMDAR Autoantibodies: Anti-GluN1

NMDARs are tetramers composed of two obligatory GluN1 subunits, associated with two regulatory subunits of GluN2-type (GluN2A-D) or GluN3-type (GluN3A-B), whose expression varies in different brain regions and developmental periods [7]. NMDARs are ligand- and voltage-gated ionotropic glutamate receptors that regulate Ca^2+^ influx. In addition, NMDAR is blocked by Mg^2+^ at resting potential, and this inhibition is usually removed by AMPA-dependent depolarization. Accordingly, NMDARs are considered a postsynaptic coincidence detector linking presynaptic glutamate release and postsynaptic depolarization [8].

Autoantibodies targeting NMDAR have been described in anti-NMDAR encephalitis (NMDARE), an autoimmune synaptic brain disorder characterized by the presence of IgG antibodies that recognize the extracellular N368/G369 region domain of the GluN1 obligatory subunit of NMDAR [9]. Adult patients affected by anti-NMDAR encephalitis present with psychosis, behavioral disorders, or confusion, as well as other symptoms such as seizures, movement disorders, sleep disorders, and irritability [10,11]. Importantly, anti-NMDAR encephalitis is considered the most common form of autoimmune encephalitis, and it accounts for about 4% of patients with any form of encephalitis [12]. The clinical features of autoimmune encephalitis are frequently preceded by symptoms suggesting an infectious process, and it has been demonstrated that a herpes simplex infection can trigger autoimmune encephalitis with neurological worsening [13]. Although viral infections might not directly trigger the disease’s onset, they may induce the breakdown of the blood–brain barrier and cause a further inflammation reaction, as well as epitope spreading and chronic activation of innate immunity actors [14]. In the same view, a few reports have suggested a link between viral SARS-COV-2 infection and anti-NMDAR encephalitis, but a definite pathogenetic role has not been postulated [15,16]. Furthermore, previous studies have suggested that vaccinations such as the H1N1 vaccine, tetanus/diphtheria/pertussis and polio vaccine, and Japanese encephalitis vaccine, might induce autoimmune encephalitis [17].

Even if early studies identified the presence of anti-GluN1 antibodies to ovarian teratoma [18], they can be associated also with non-paraneoplastic forms and other causes including viral infections. Moreover, although some studies reported the presence of anti-NMDAR IgM and IgA in about 10% of patients affected by schizophrenia, the same type of antibodies were also identified in patients with other brain disorders, including dementia, stroke, and Parkinson’s disease, thus suggesting the absence of specificity and, therefore, a reduced clinical relevance [19,20,21]. Moreover, conversely to what was clearly demonstrated for anti-NMDAR IgG found in anti-NMDAR encephalitis, it seems that anti-NMDAR IgM and IgA associated with other diseases do not alter NMDAR synaptic function [22]. Finally, between 4% and 7.5% of patients with anti-NMDAR encephalitis have concurrent glial or neuronal-surface antibodies, with the most frequent association being with myelin oligodendrocyte glycoprotein (MOG) or aquaporin 4 (AQP4) antibodies, which result in overlapping anti-NMDAR encephalitis with demyelinating disorders [23,24]. The presence of these autoimmune associations has clinical implications, conferring additional clinical and radiologic features to anti-NMDAR encephalitis and influencing prognosis and treatment approaches [23].

Since the first report in 2007 [18], several studies have been performed not only related to the clinical management of the disease, but also to address the precise molecular mechanisms by which the presence of anti-GluN1 antibodies induces pathological alterations of the glutamatergic neurotransmission [3,25]. Early in vitro studies showed that the incubation of primary hippocampal neurons with anti-GluN1 antibodies from patients was sufficient to significantly decrease GluN1 or NMDAR surface levels and synaptic clusters in a titer-dependent fashion compared to controls [26] (Figure 1). In addition, GluN1 autoantibodies weakened the interaction between NMDARs and ephrin-B2 receptors (EPHB2R), and the antibody-dependent reduction of NMDAR clusters was prevented both in vitro and in vivo by the activation of EPHB2R [27]. Interestingly, the decreased interaction with EPHB2R altered the dynamics of synaptic and extrasynaptic NMDARs, possibly leading to an increased lateral diffusion toward extrasynaptic sites where the receptor can be internalized [28]. However, a direct action of the anti-GluN1 antibodies on the extrasynaptic pool of NMDARs leading to fast receptor internalization and/or a decrease in receptor diffusion and synaptic incorporation has also been postulated [5]. The effects on NMDAR were reversible after anti-GluN1 antibody removal from cultures and were not associated with alterations of other AMPAR and PSD-95 clusters, or with morphological events such as reduced branching or spine loss. Finally, anti-GluN1 antibodies were not associated in vitro with modifications of neuronal viability [26]. Consistent with this molecular evidence, the patients’ antibodies specifically decreased synaptic NMDAR-mediated currents and did not affect AMPAR-mediated currents. Electrophysiological analysis further supports the idea that a decrease in synaptic NMDAR-mediated currents was associated with the low level of NMDARs induced by autoantibodies, rather than a direct antibody blockade [26,28]. These alterations at the NMDAR level also lead to reduced synaptic plasticity, as indicated by profound modifications of molecular signatures of long-term potentiation (LTP) [27]. Moreover, treatment with the patients’ antibodies did not affect miniature excitatory postsynaptic currents (mEPSC) frequency or amplitude of hippocampal neurons, thus indicating the absence of any effect on presynaptic release probability [26].

In agreement with in vitro data, in vivo chronic infusion of anti-GluN1 antibodies into a rat hippocampus led to a significant decrease in NMDAR cluster density and intensity of GluN1 immunostaining without affecting the number of synapses, the density of other synaptic components, or neuronal death, and always in a titer-dependent manner [26]. From a kinetic point of view, surface NMDAR cluster density was significantly decreased 12 h after exposure to patient antibodies [28]. Related to this, similar molecular alterations of NMDAR staining and cluster density were observed in paraffin-embedded hippocampal sections from patients with anti-NMDAR encephalitis, thus providing a key indication that the use of autoantibodies in in vitro neuronal cultures and in vivo murine models can reproduce molecular alterations present in the brain of patients with the disorder.

More recent studies have evaluated the behavioral outcome after chronic infusion of autoantibodies in mice [29]. Animals infused with patients’ cerebrospinal fluid (CSF) displayed a progressive worsening of memory deficits and depressive-like behaviors, but with no effect on other social or locomotor skills. Interestingly, similar to the above-mentioned molecular events, memory deficits were also progressive and fully reversible one week after the end of the infusion [29], thus demonstrating the existence of a direct link between behavioral deficits and antibody-induced effects on NMDARs. Notably, in vivo treatment of ephrin-B2 prevented memory impairments and depressive-like behavior, as well as molecular alterations of synaptic NMDAR and impairments of synaptic plasticity [30].

As mentioned above, recent studies have addressed a possible role for anti-NMDAR not only in encephalitis, but also in other brain disorders. Jezequel and colleagues investigated the impact of anti-NMDAR IgG purified from a cohort of patients with schizophrenia and matched healthy subjects, excluding cases of autoimmune encephalitis [31]. Antibodies against NMDAR were observed in about 20% of psychotic patients diagnosed with schizophrenia. These antibodies induced in vitro destabilization of synaptic NMDAR and its anchoring partner EPHB2R. Consequently, anti-NMDAR from psychotic patients decreased the NMDAR synaptic content and impaired induction NMDAR-dependent long-term potentiation (LTP) [31].

A recent study evaluated the in vitro effect of anti-NMDAR antibodies on the activity of oligodendrocyte NMDARs by using a functional assay based on cytosolic calcium imaging [32]. Preincubation of oligodendrocyte culture with encephalopathy patients’ CSF containing the antibody induced a significant reduction of NMDAR responses and, similar to previous reports for neuronal cells, AMPAR function remained largely unaltered. Interestingly, the patients’ CSF also reduced oligodendrocyte expression of glucose transporter GLUT1 induced by NMDAR activity. Notably, this study theorized the existence of a possible link between antibody-mediated dysfunction of NMDARs in oligodendrocytes and the white-matter alterations reported in patients [33].

Even if the autoimmune nature of anti-NMDAR encephalitis involves the presence of inflammatory infiltrates identified in postmortem brain specimens [34], the immune and inflammatory events that play a key role for this pathogenic process are still far from being clearly understood. Importantly, a recent study that used an active immunization approach to address this issue demonstrated that after two weeks, mice developed clinical symptoms reminiscent of encephalitis, including anxiety- and depressive-like behaviors, a deficit in spatial memory, and increased sensitivity to seizures. This response was associated with B-cell infiltration toward the ventricles, where they differentiated into plasmacytes [35]. Notably, the use of depleting antibodies to block the B-cell response induced a reduction of the severity of symptoms in the animal model [35].

## 3. NMDAR Autoantibodies: Anti-GluN2B

Different studies demonstrated the presence of anti-NMDA antibodies against other subunits of NMDARs, mainly in patients with epilepsy. In an initial study, anti-GluN2B antibodies were detected in the serum in 13 of 15 patients with chronic epilepsia partialis continua, including patients with histologically proven Rasmussen’s encephalitis, but not in patients with West syndrome or Lennox–Gastaut syndrome [36]. The antibody recognized predominantly GluN2B C-terminal epitopes and rarely the N-terminal epitope. More recently, CSF levels of anti-GluN2B antibodies raised against both N-terminal and C-terminal epitopes were higher in Rasmussen’s syndrome patients than in disease controls. The antibodies, which increased to the highest levels from 12 to 23 months after epilepsy onset, were associated with seizure aggravation [37]. Similarly, Mori and coworkers [38] found higher CSF levels of antibodies to the GluN2B subunit in patients with epileptic encephalopathy, also known as West syndrome or epilepsy with epileptic spasms. These GluN2B antibodies are detected after the onset of epileptic spasms, and probably are associated with neural damage. Moreover, in an attempt to identify shared underlying mechanisms between systemic lupus erythematosus (SLE) and fibromyalgia (FM), Park and colleagues [39] found anti-GluN2B antibody in patients with SLE, and those patients with concomitant FM had higher anti-GluN2B antibody titers. Finally, a case report described a patient who developed refractory schizophrenia that mimicked an exacerbation of encephalitis with antibodies to GluN2B after administration of clozapine [40], thus suggesting the importance of differentiating between a diagnosis of refractory schizophrenia and encephalitis with antibodies to GluN2B.

## 4. AMPAR Autoantibodies: Anti-GluA1 and Anti-GluA2

AMPARs are tetrameric ligand-gated ion channels composed of GluA1-4 subunits that play a key role in regulating fast excitatory glutamatergic neurotransmission. Excitatory synapses mainly express AMPAR channels composed by GluA1/GluA2 or GluA2/GluA3 [41]. Importantly, AMPARs’ function, including kinetics, ligand pharmacology, and membrane retention, are strictly modulated by their subunit composition [41].

Autoantibodies that target AMPAR subunits, both GluA1 and GluA2, have been found in patients with autoimmune encephalitis [42,43,44,45]. Typically, these patients show limbic encephalitis with the presence of short-term memory loss, confusion, mood disturbances, epilepsy, and sleep disorders, but other symptoms or psychosis may occur, and anti-AMPAR encephalitis is paraneoplastic in 64% of cases [11,43,46].

Treatment of primary rat hippocampal neurons with GluA1-GluA2 autoantibodies induced a significant decrease in the synaptic levels of AMPAR subunits that was promptly reversed after antibody removal [43,45]. However, the patients’ antibodies did not modify NMDAR clusters, dendritic spine density, or cell survival [45]. In addition, electrophysiological analysis showed that the patients’ antibodies reduced AMPAR- but not NMDAR-mediated currents [45]. A more recent study confirmed that GluA2 autoantibodies promote receptor internalization and a consequent decrease in synaptic GluA2-containing AMPARs [47]. Interestingly, this event was followed by a compensatory ryanodine receptor-dependent insertion at synapse of AMPARs not containing the GluA2 subunit. Notably, treatment with anti-GluA2 antibodies in mice impairs long-term potentiation and induce recognition memory deficits and anxiety-like behavior [47].

## 5. AMPAR Autoantibodies: Anti-GluA3

Initial studies described the presence in the serum of an autoantibody recognizing the GluA3 subunit of AMPAR in Rasmussen’s encephalitis and in up to 20% to 30% of patients affected by different types of epilepsy [4,48,49,50,51]. Interestingly, the presence of anti-GluA3 antibodies was correlated with the presence of learning, attention, and psychiatric problems [50]. An analysis of the molecular and cellular effects induced by anti-GluA3 antibodies revealed that they can bind GluA3-containing AMPAR on neurons, probably acutely inducing receptor activation, despite the interaction with a different site than glutamate, thus suggesting a possible involvement in excitotoxicity [4,52].

Recently, the identification of autoantibodies against the GluA3 subunit of AMPAR in about 20% to 25% of patients with frontotemporal dementia (FTD) [53] opened new indications for a pathogenetic role of glutamate receptors’ autoantibodies in neurodegenerative diseases characterized by alterations of glutamatergic neurotransmission [54,55]. FTD, a common cause of presenile dementia, is a clinically heterogeneous disorder characterized at the onset by behavioral abnormalities, personality changes, deficits of executive functions, and language impairment [56,57]. Clinically, FTD encompasses three clinical syndromes: behavioral variant FTD (bvFTD), the agrammatic variant of primary progressive aphasia (avPPA), and the semantic variant of PPA (svPPA) [58,59]. The neuropathological substrate is heterogeneous, with hyperphosphorylated Tau or transactive response DNA-binding protein 43 (TDP-43) being the most frequent underlying proteinopathies, responsible for frontotemporal lobar degeneration (FTLD)-tau or FTLD-TDP43, respectively [60].

A dysregulation of the immune system, at the level of both the inflammatory and the autoimmune component, has recently been proposed to contribute to the pathogenesis and neurodegeneration of FTD and a crosstalk between autoimmunity and altered glutamatergic neurotransmission in triggering the disease’s onset [55,61]. This hypothesis stemmed from epidemiological data and clinical observations, which reported a significantly increased risk of autoimmune disorders and autoimmune system dysregulation in FTD patients, as well as from genetic research studies that argued for immune-mediated genetic enrichment in FTD [62,63,64,65].

The recent studies that reported a high frequency of autoantibodies recognizing the GluA3 subunit of AMPAR (anti-GluA3 Abs) in patients with FTD [1,53] also corroborated the evidence for altered glutamatergic transmission in this disorder [55,66]. GluA3 is a highly relevant subunit of AMPAR in the brain, contained in a high proportion of cortical AMPAR [67]. From a functional point of view, GluA2/GluA3 AMPAR are recruited in a constitutive manner to synapses, where they can replace GluA1-containing receptors that are usually added to synaptic membranes during plasticity [68,69].

Early in vitro studies showed that acute treatment with anti-GluA3 antibodies is detrimental to AMPAR function and neuronal viability (Figure 2) [53]. In particular, incubation of rat hippocampal neuronal primary cultures with patients’ CSF containing anti-GluA3 antibodies led to a decrease in GluA3 subunit synaptic localization and a loss of dendritic spines [53]. These results were further confirmed in differentiated neurons from human-induced pluripotent stem cells, with a significant reduction of the GluA3 subunit in the postsynaptic fraction [53]. A more recent study showed that anti-GluA3 antibodies decreased AMPA-evoked glutamate exocytosis from synaptosomes in a dose-dependent manner [1]. Accordingly, it is possible to affirm that the presence of GluA3 antibodies can impact glutamatergic neurotransmission acting at the presynaptic terminal by reducing glutamate release, and on dendritic spines by lowering the availability in the postsynaptic membranes of AMPAR.

An analysis of the effects induced by GluA3 antibodies in postmortem specimens of the frontotemporal and occipital cortex of FTLD-tau patients demonstrated that subjects carrying the GluA3 antibodies are characterized by a significant reduction in GluA3 levels in a postsynaptic fraction purified from the frontotemporal cortex, whereas no differences were detected in the postsynaptic levels of other AMPAR subunits [1]. A possible explanation for different synaptic retention of AMPARs could be found in protein-protein interaction, which plays a key role in these events [59]. Two scaffolding proteins, GRIP1 and PICK1, are mainly involved in receptor insertion/internalization at the postsynaptic membrane [59]. Accordingly, the reduced GluA3 localization at synapses in anti-GluA3-positive FTLD-tau patients is accompanied by a significant increased PICK1/GRIP1 ratio, thus suggesting that GluA3 Abs promote endocytosis of the receptor subunit, probably through interaction with PICK1 [1] (Figure 2). No alterations of the synaptic and total levels of NMDAR subunits were observed in anti-GluA3-positive FTD-tau patients, thus suggesting that GluA3 antibodies target AMPAR without interfering with other glutamate receptor subtypes [1].

Interestingly, the use of neurophysiological techniques to perform an in vivo assessment of excitatory glutamatergic circuits in patients revealed a significant difference in intracortical facilitation (ICF) between those positive for GluA3-antibodies and those negative, with a significantly increased glutamatergic deficit in the former group [1]. Interestingly, ICF was previously shown to be deficient in both sporadic and genetic FTD patients compared to healthy controls [70], and to correlate with disease progression [71]. Finally, antibody titer seems to correlate with age at the disease’s onset, with earlier onset of symptoms observed in those FTD patients with higher antibody levels [53].

Overall, the presence of anti-GluA3 Abs in FTD patients corroborates other evidence claiming a possible role of AMPARs in FTD pathogenesis: The hyperexcitability of AMPARs contributed to neurodegeneration [72]; social deficits were accompanied by a change in AMPAR composition in an animal model of FTD [72]; frontal cortex and human-induced pluripotent stem cells (hiPSCs) of behavioral-variant FTD patients showed changes in AMPARs [73]; and the physiological release of Tau protein was mediated by AMPARs [74]. Although this evidence points to a role for anti-GluA3 Abs in FTD, the mechanisms underlying the functional effects on AMPARs are still not completely understood. Further studies are needed to assess the functional consequences of anti-GluA3 Abs on cognitive functions and behavior. Finally, the identification of the molecular basis of immune-system dysregulation in FTD might open new routes for therapeutic perspectives in autoimmune-related neurodegeneration, which can help reduce or revert the disease’s progression.

## 6. Conclusions

Pharmacological modulation of the immune response is the gold-standard treatment for anti-NMDAR encephalitis that leads to a substantial improvement of symptoms in the majority of patients [75,76]. However, there is general agreement regarding the need for additional therapies for brain diseases characterized by the presence of anti-NMDAR or anti-AMPAR antibodies.

Preclinical studies performed in the last decade allowed for a detailed characterization of the molecular, functional, and morphological alterations induced by autoantibodies at the glutamatergic synapses in both in vitro and in vivo models, and it is now possible to design novel treatments that specifically target glutamate receptors. A recent report from Warikoo and coworkers [77], which characterized the effects in vitro of recently described positive allosteric modulators (PAMs) of NMDARs, showed that co-application of NMDAR PAMs with patients’ CSF carrying anti-NMDAR antibodies restored NMDAR function. Further preclinical in vivo studies with NMDAR PAMs are needed and may lead to new treatments for anti-NMDAR encephalitis and other brain disease characterized by reduced activity of synaptic NMDAR.

Regarding anti-AMPAR antibodies, FTD patients that carry anti-GluA3 antibodies are characterized by a reduction of both the postsynaptic expression of GluA3-containing AMPA receptors and a decreased intracortical facilitation, which predicts a more pronounced impairment of glutamatergic neurotransmission. Considering the key role of AMPA receptors in the regulation of glutamatergic neurotransmission, the recent availability of AMPA PAMs already approved for clinical use or currently under evaluation [78,79,80,81] could represent an intriguing approach to rescue a physiological excitatory synaptic transmission in FTD patients.

In conclusion, a giant step forward has been taken in recent years in the understanding of the molecular bases of glutamatergic synapse autoimmunity and the related clinical features. However, it is possible that the acute onset of autoimmune encephalitis described so far is just the tip of the iceberg, and that subacute autoimmune processes may alternatively lead to subtle and progressive clinical changes. Future studies to establish the role of autoantibodies directed against subunits of both NMDAR and AMPAR in chronic cognitive and behavioral disorders should be further addressed.

## Figures and Tables

**Figure 1 cells-10-00077-f001:**
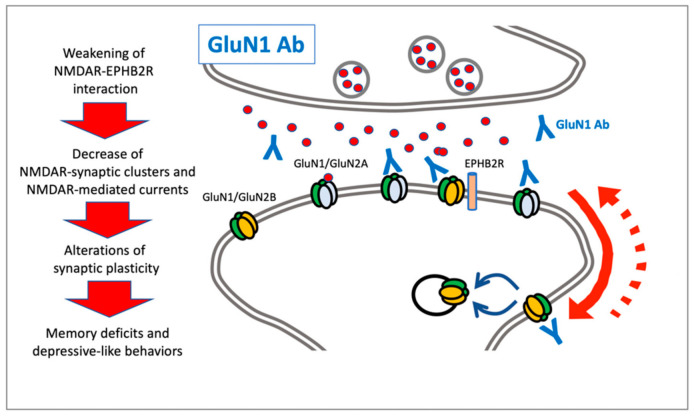
Graphic representation of molecular and cellular mechanisms triggered by anti-GluN1 antibodies through interaction with synaptic or extrasynaptic NMDARs.

**Figure 2 cells-10-00077-f002:**
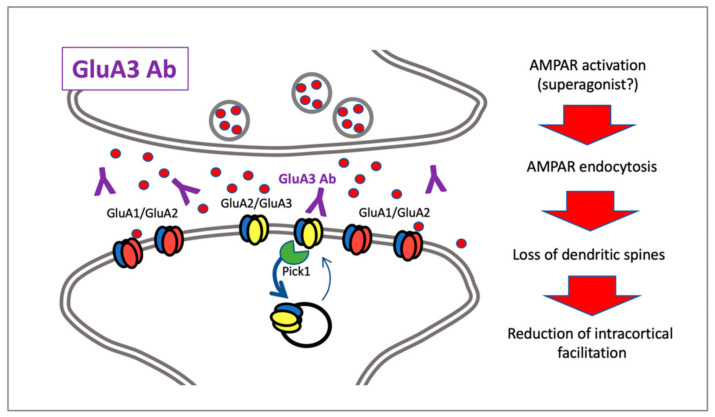
Graphic representation of molecular and cellular mechanisms triggered by anti-GluA3 antibodies.

**Table 1 cells-10-00077-t001:** Clinical and preclinical effects of NMDAR and AMPAR autoantibodies.

Ionotropic Glutamate Receptor Subunit	GluN1	GluN2B	GluA1	GluA2	GluA3
Main clinical syndromes	Anti-NMDAR encephalitis	Rasmussen encephalitis, West syndrome	Limbic encephalitis, epilepsy		Rasmussen’s encephalitis, epilepsy, frontotemporal dementia
Main clinical symptoms	Memory and behavioral disturbances, seizures, psychosis, sleep disorders	Seizures	Cognitive deficits, anxiety, sleep disturbances, mood disturbances and epilepsy		Cognitive and psychiatric symptoms, seizures
Molecular-functional effects (in vitro)	Decreased NMDAR synaptic clusters and currents	Mostly unknown	Decreased AMPAR synaptic levels	Decreased AMPAR synaptic levels and currents	AMPAR internalization, spine loss
Molecular-functional effects (animal models; ex vivo)	Synaptic plasticity impairment, memory deficits, depressive-like behavior	Mostly unknown	Mostly unknown	AMPAR internalization, synaptic plasticity impairment, memory deficits, depressive-like behavior	AMPAR internalization, reduced intracortical facilitation

## Data Availability

Not applicable.

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
