# Peer review of "NMDA and AMPA Receptor Autoantibodies in Brain Disorders: From Molecular Mechanisms to Clinical Features"

_cells, 2021, doi:10.3390/cells10010077_

Round 1

Reviewer 1 Report

The author performed a literature study of autoimmunity against the subunits of ionotropic glutamate receptors NMDA receptors (NMDAR) and AMPA receptor (AMPAR) subunits which caused corresponding CNS disorders. The author summarized the molecular function of each subunit, clinical manifestation, mechanism,  experiment finding, and possible correlation with other CNS disorders. 

page 3 second paragraph: When the authors describe the mechanism of GluN1  decreased, what mechanism of the GluN1 loss from the membrane?  Do authors consider this is an antigen modulation process/autoantibody-mediated antigen internalization?  

Similarly, the authors reflected the AMPAR endocytosis in figure 2 but didn't show how the NMDAR decrease happened in figure 1.  If the authors consider the different mechanisms between the two receptors, the authors should point it out clearly. 

Author Response

REVIEWER: The author performed a literature study of autoimmunity against the subunits of ionotropic glutamate receptors NMDA receptors (NMDAR) and AMPA receptor (AMPAR) subunits which caused corresponding CNS disorders. The author summarized the molecular function of each subunit, clinical manifestation, mechanism,  experiment finding, and possible correlation with other CNS disorders. 

page 3 second paragraph: When the authors describe the mechanism of GluN1  decreased, what mechanism of the GluN1 loss from the membrane?  Do authors consider this is an antigen modulation process/autoantibody-mediated antigen internalization?  

REPLY: We agree with the Reviewer that both the text and Figure 1 were incomplete because they did not take into consideration antibody-dependent receptor internalization as well as a possible direct activity of the anti-GluN1 antibodies on extrasynaptic NMDA receptors. We have now added these putative mechanisms induced by anti-GluN1 antibodies in the revised version of the text and Figure 1 (lines 112-117).

REVIEWER: Similarly, the authors reflected the AMPAR endocytosis in figure 2 but didn't show how the NMDAR decrease happened in figure 1.  If the authors consider the different mechanisms between the two receptors, the authors should point it out clearly. 

REPLY: In the text of the revised version of the review, we have added a sentence indicating that the presence of anti-GluA3 antibodies in FTD patients do not interfere with NMDARs levels (lines 538-541). About the mechanisms described in the figures, differently from Figure 1, we decided in Figure 2 to focus our attention on receptor internalization because no information is at present available on antibody-dependent alterations of AMPA receptor movement between synaptic and extrasynaptic pools.

Reviewer 2 Report

In this review manuscript (cells-1059494), Gardoni and colleagues summarized the role of NMDA and AMPA receptor autoantibodies in brain disorders. Some comments and suggestions are listed as below:

  1. A similar review on this topic has already been published (Hunter et al, Autoimmunity and NMDA receptor in brain disorders: Where do we stand? Neurobiology of Disease, 2021).
  2. Please keep the font consistent in the manuscript (Lines 32-44). The same problem was also noted in lines 303-310.
  3. When it comes to disease, the authors used NMDAR encephalitis (NMDARE) in line 58, while anti-NMDAR encephalitis was used in line 108. I prefer using anti-NMDA receptor (NMDAR) encephalitis (See Dalmau J, et al. Anti-NMDA-receptor encephalitis: case series and analysis of the effects of antibodies. Lancet Neurol 2008;7:1091-8; Dalmau J, et al. Clinical experience and laboratory investigations in patients with anti-NMDAR encephalitis. Lancet Neurol 2011;10:63-74; Lim JA, et al. Anti-N-methyl-d-aspartate receptor encephalitis in Korea: clinical features, treatment, and outcome. J Clin Neurol 2014;10:157-61). Please double check.
  4. The authors said that anti-NMDA receptor encephalitis can be associated with viral infections in lines 66-67. This is an interesting point. Although viral infections might not directly trigger the disease onset, it may induce the breakdown of the blood brain barrier and cause a further inflammation reaction. Please discuss potential mechanisms. Do you think a systematic search for viral triggers is necessary?
  5. How about the association between SARS-CoV-2 infection and the development of anti-NMDA receptor encephalitis? How about the relationship between vaccines and anti-NMDA receptor encephalitis (NMDAR autoantibodies)?
  6. In Figure 1, ‘memory deficits’ should be ‘Memory deficits’. Remove the light red line of behaviors.
  7. What do you mean by saying ‘see below’ in line 72?
  8. The format of ref. 4 is wrong.
  9. It was reported that 4%-7.5% patients with anti-NMDA receptor encephalitis may experience overlapping with other autoimmune diseases originating from glial or neuronal surface antigens. For example, clinical cases of coexisting anti-NMDAR encephalitis and MOG antibody-associated encephalomyelitis have been reported recently. Potential mechanisms should be discussed in the revised manuscript.
  10. In the part of NMDAR autoantibodies: anti-GluN2B, a possible relationship between NMDA receptor antibodies, refractory schizophrenia, and clozapine has been reported (PMID: 27482955), which was not mentioned by the authors.
  11. Rasmussen’s encephalitis (RE) was used in line 191, while Rasmussen’s encephalitis was used in line 153.
  12. The title of this review NMDA and AMPA receptor autoantibodies in brain disorders. Sleep disorders should also be discussed since autoimmune encephalitis assocaited with anti-NMDA and AMPA receptor autoantibodies can affect the brain network involved in sleep initiation and regulation.

Author Response

In this review manuscript (cells-1059494), Gardoni and colleagues summarized the role of NMDA and AMPA receptor autoantibodies in brain disorders. Some comments and suggestions are listed as below.

  1. A similar review on this topic has already been published (Hunter et al, Autoimmunity and NMDA receptor in brain disorders: Where do we stand? Neurobiology of Disease, 2021).

REPLY: We want to thank the Reviewer for informing us about the publication of this review that has been added in the reference list (Ref. #5). However, it is important to point out that the very recent review of Hunter and colleagues focuses on NMDAR autoantibodies and does not address the role of AMPAR autoantibodies.

  1. Please keep the font consistent in the manuscript (Lines 32-44). The same problem was also noted in lines 303-310.

REPLY: We want to thank the Reviewer for this mistake probably due to the copy-paste of the text in the journal template. We have now kept the font consistent throughout the manuscript.

  1. When it comes to disease, the authors used NMDAR encephalitis (NMDARE) in line 58, while anti-NMDAR encephalitis was used in line 108. I prefer using anti-NMDA receptor (NMDAR) encephalitis (See Dalmau J, et al. Anti-NMDA-receptor encephalitis: case series and analysis of the effects of antibodies. Lancet Neurol 2008;7:1091-8; Dalmau J, et al. Clinical experience and laboratory investigations in patients with anti-NMDAR encephalitis. Lancet Neurol 2011;10:63-74; Lim JA, et al. Anti-N-methyl-d-aspartate receptor encephalitis in Korea: clinical features, treatment, and outcome. J Clin Neurol 2014;10:157-61). Please double check.

REPLY: We agree with the Reviewer on this point and we have revised the text accordingly.

  1. The authors said that anti-NMDA receptor encephalitis can be associated with viral infections in lines 66-67. This is an interesting point. Although viral infections might not directly trigger the disease onset, it may induce the breakdown of the blood brain barrier and cause a further inflammation reaction. Please discuss potential mechanisms. Do you think a systematic search for viral triggers is necessary?

REPLY: As correctly requested, we expanded the discussion on the association between anti-NMDA receptor encephalitis and viral infections. In the revised version, we added: “Indeed, the clinical features of autoimmune encephalitis are frequently preceded by symptoms suggesting an infectious process, and it has been demonstrated that herpex simplex infection can trigger autoimmune encephalitis with neurological worsening (Armangue T et al., Lancet Neurol 2018). Although viral infections might not directly trigger the disease onset, this may induce the breakdown of the blood brain barrier and cause a further inflammation reaction as well as epitope spreading and chronic activation of innate immunity actors (Joubert B and Dalmau J. Rev Neurol (Paris) 2019;175:420-426)” (see lines 68-73).

Indeed, a systematic search for viral triggers may be unnecessary when an autoimmune encephalitis is suspected, as the viral infection may precede the development of autoimmune encephalitis and the possible spectrum of viral causes may be too wide and still unknown.

  1. How about the association between SARS-CoV-2 infection and the development of anti-NMDA receptor encephalitis? How about the relationship between vaccines and anti-NMDA receptor encephalitis (NMDAR autoantibodies)?

REPLY: In the same view, as reported above, some reports have suggested a link between SARS-CoV-2 infection and anti-NMDAR encephalitis, but the available data are still too scarce to allow a definitive conclusion. Moreover, as correctly pointed out, previous studies claimed that vaccination might induce autoimmune encephalitis. In the revised version of the manuscript, we specified these points and we added the available literature references: “In the same view, a few reports have suggested a link between viral SARS-COV-2 infection and anti-NMDAR encephalitis, but a definite pathogenetic role may not be postulated (Monti et al., Seizure 2020; Panariello et al., Brain Behav Immnunity 2020). Furthermore, previous studies have suggested that vaccination, such as H1N1 vaccine, tetanus/diphtheria/pertussis and polio vaccine, and Japanese encephalitis vaccine, might induce autoimmune encephalitis (Wang H, Curr Pharm Des 2020) (see lines 73-86).

  1. In Figure 1, ‘memory deficits’ should be ‘Memory deficits’. Remove the light red line of behaviors.

REPLY: We have modified Figure 1 as indicated by the Reviewer.

  1. What do you mean by saying ‘see below’ in line 72?

REPLY: We have deleted “see below” from the revised text.

  1. The format of ref. 4 is wrong.

REPLY: Even if we fully agree with the Reviewer that the mentioned reference seems wrong, this is actually the reference format reported in Pubmed and used in other papers mentioning this reference.

  1. It was reported that 4%-7.5% patients with anti-NMDA receptor encephalitis may experience overlapping with other autoimmune diseases originating from glial or neuronal surface antigens. For example, clinical cases of coexisting anti-NMDAR encephalitis and MOG antibody-associated encephalomyelitis have been reported recently. Potential mechanisms should be discussed in the revised manuscript.

REPLY: As correctly pointed out, overlapping autoimmune disorders may occur along with anti-NMDAR encephalitis, and in particular with demyelinating diseases, and we clarified this issue in the revised version: “…Finally, between 4% and 7.5% of patients with anti-NMDARE have concurrent glial or neuronal-surface antibodies, the most frequent association being with myelin oligodendrocyte glycoprotein (MOG) or aquaporin 4 (AQP4) antibodies, which result in overlapping anti-NMDAR encephalitis with demyelinating disorders (Martinez-Hernandez E et al., Neurology 2020; Titulaer MJ et al., Ann Neurol 2014). The presence of these autoimmune associations has clinical implications, conferring additional clinical and radiologic features to anti-NMDAR encephalitis and influencing prognosis and treatment approaches (Martinez-Hernandez E et al., Neurology 2020) (see lines 96-102).

  • In the part of NMDAR autoantibodies: anti-GluN2B, a possible relationship between NMDA receptor antibodies, refractory schizophrenia, and clozapine has been reported (PMID: 27482955), which was not mentioned by the authors.

REPLY: We want to thank the Reviewer for raising this issue. We have now expanded the section related to anti-GluN2B autoantibodies describing the work by Gon and colleagues (see lines 271-275).  

  • Rasmussen’s encephalitis (RE) was used in line 191, while Rasmussen’s encephalitis was used in line 153.

REPLY: We have corrected this issue as correctly suggested by the Reviewer.

  • The title of this review NMDA and AMPA receptor autoantibodies in brain disorders. Sleep disorders should also be discussed since autoimmune encephalitis assocaited with anti-NMDA and AMPA receptor autoantibodies can affect the brain network involved in sleep initiation and regulation.

REPLY: We agree with the Reviewer, and we added this interesting issue on sleep disturbances in the text of the manuscript and in the figure as well as the related reference.

Round 2

Reviewer 2 Report

The authors have addressed my concerns. I have one minor concern. In line 68 of revised manuscript, the authors said that herpex simplex infection can trigger autoimmune encephalitis with neurological worsening. However, 'herpex simplex infection' should be 'herpes simplex infection'. Please double check.